# XDex: Learning Cross-Embodiment Dexterous Grasping with 1000 Hands

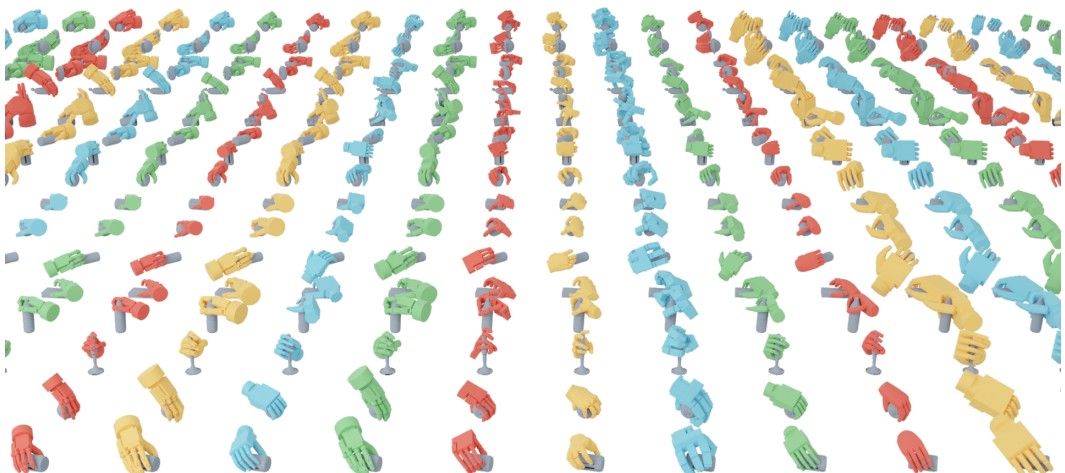

Figure 1: **XDex** learns to synthesize consistent and diverse grasps across embodiments. Each row shows the same grasping pose for the same object across different robot hands. Each column corresponds to a single robot hand (color-coded) grasping different objects.

## Abstract

Synthesizing dexterous grasps across various hands remains a fundamental challenge in robotic manipulation due to morphology gaps in geometry, topology, and kinematics. We hypothesize that scaling the diversity and number of hand embodiments improves generalization to unseen hands. To this end, we introduce XDex, a framework trained on the largest cross embodiment grasping dataset, which we built using 1,000 diverse hands. XDex features an embodiment transformer that jointly encodes hand geometry and topology to learn from this large scale dataset. Additionally, we enforce grasp consistency across embodiments by training on a paired grasping dataset and introducing a retargeting loss. The paired data are generated by first synthesizing grasps for a source hand and then translating them to diverse target hands. XDex significantly outperforms prior methods in grasp quality, consistency, and diversity, and demonstrates strong generalization to unseen hands in real world settings. We show more qualitative results anonymously at `https://Xdex-ICLR.github.io`

## 1 Introduction

Dexterous grasping is a fundamental yet challenging task in robotics, made even more complex when generalizing across embodiments. A grasp pose that works for one hand (e.g., a 27-DOF human hand) may fail when retargeted to another (e.g., a 6-DOF Ability Hand) due to morphology gaps in terms of geometry, topology, and kinematics. Traditional learning-based grasp synthesis methods often target a specific hand, limiting their generalization to new hands.

Researchers have explored generalization across embodiments for years, with existing approaches falling into two main categories. One line of work Shao et al. (2020); Attarian et al. (2023); Li et al. (2023); Xu et al. (2024b); Morrison et al. (2018); Varley et al. (2015) utilizes object-centric

representations, such as contact points or contact maps, followed by inverse kinematics to infer joint values. While these methods enable generalization, they often suffer from limited performance due to a lack of understanding of robot hand morphology. Another line of work Wang et al. (2018); Attarian et al. (2023); Sferrazza et al. (2024); Patel & Song (2024); Wei et al. (2025) conditions the model on robot hand descriptions, but still struggles to achieve strong performance on unseen hands, since they are trained with only a limited number of hands and cannot fully capture the diversity of hand morphology.

Inspired by scaling laws in computer vision Kirillov et al. (2023); Oquab et al. (2023); Caron et al. (2021) and natural language processing Kaplan et al. (2020); Chowdhery et al. (2023); Achiam et al. (2023), we hypothesize that scaling the diversity and number of hand embodiments leads to better grasping performance on unseen hands. We then construct the **largest cross-embodiment grasping dataset** with 1,000 hands covering diverse geometries and topologies and introduce **XDex**, a novel framework for cross-embodiment dexterous grasp synthesis. Specifically, the dataset contains procedurally generated robotic hands with varied link shapes and articulation structures, as well as human hands of different shapes, together providing broad diversity in geometry and topology. To enable learning from large-scale embodiments, we introduce an embodiment transformer encoder that jointly encodes hand geometry and topology. The geometry is captured with per-link point encoders, while the topology is represented through an attention mask that encodes joint connectivity.

Beyond embodiment scaling, we also enforce consistency in the grasping poses across different hands. This not only enables new grasp-to-grasp translation applications (for example, object-conditioned human-to-robot neural retargeting) but also improves policy learning. The model is conditioned on the object's local geometry features and the hand, and it must produce consistent grasping poses under identical conditions. In this way, the model transfers a grasp to a novel hand with minimal adjustment. To achieve this, we construct a paired grasping dataset to teach the model consistency across hands. Specifically, we first synthesize high-quality grasps for a source human hand using force-closure optimization, and then translate these poses to diverse target hands through multiple objectives. The model is trained on this paired dataset to implicitly learn consistency. In addition, we introduce a *retargeting loss* that explicitly enforces consistency in the predictions.

We evaluate XDex on a large-scale benchmark covering both seen and unseen hands, and report results across grasp quality, consistency, and diversity. Our method significantly outperforms prior approaches and demonstrates strong generalization to unseen hands on a real-world robotic platform.

In summary, our contributions are three-fold:

- We propose XDex, a framework that learns cross-embodiment grasping by encoding hand geometry and topology and enforcing grasp consistency across hands.
- We build the largest cross-embodiment grasping dataset with 1,000 diverse hands and paired grasps for learning grasp transfer.
- We conduct extensive experiments, including real-world deployment, to validate the effectiveness and scalability of our approach.

## 2 RELATED WORK

**Dexterous Grasping.** Dexterous grasping has been extensively studied in the robotics community for decades. Classical methods Miller & Allen (2004); Ciocarlie et al. (2007); Ferrari et al. (1992); Bai & Liu (2014) typically involve maximizing analytic grasp metrics through optimization or sampling. However, these methods often perform poorly on high-DoF hands and require ground-truth object models. In recent years, learning-based approaches—both reinforcement learning (RL) and imitation learning (IL)—have shown promising results. On the IL side, a number of works Jiang et al. (2021); Lundell et al. (2021); Shao et al. (2020); Li et al. (2023); Ye et al. (2023); Lu et al. (2024); Weng et al. (2024); Liu et al. (2024); Xu et al. (2024a); Lum et al. (2024a); Ye et al. (2025) leverage large-scale grasp datasets to train grasp synthesis models. Datasets are collected using different approaches: human demonstrations in ContactDB Brahmbhatt et al. (2019) and DexYCB Chao et al. (2021), force-closure optimization in DexGraspNet Wang et al. (2023); Zhang et al. (2024b), and differentiable simulation in Grasp'D Turpin et al. (2022; 2023). On the RL side, UniDexGrasp Xu et al. (2023); Wan et al. (2023) and DexPoint Qin et al. (2023a) learn dexterous grasping policies from point clouds. GraspXL Zhang et al. (2024a) trains an RL policy

| Method | Grasp Representation | Cross Embodiment | Consistent Grasping Poses | Inference Speed | Retargeting Application | Full-hand Contact |
|---|---|---|---|---|---|---|
| UniDexGrasp++ Wan et al. (2023) | Joint Values | ✗ | ✗ | ✓ | ✗ | ✓ |
| UniGrasp Shao et al. (2020) | Contact Point | ✓ | ✗ | ✗ | ✗ | ✗ |
| GeoMatch Attarian et al. (2023) | Contact Point | ✓ | ✗ | ✗ | ✗ | ✓ |
| GenDexGrasp Li et al. (2023) | Contact Map | ✓ | ✗ | ✗ | ✗ | ✓ |
| ManiFM Xu et al. (2024b) | Contact Map | ✓ | ✗ | ✗ | ✗ | ✗ |
| DRO-Grasp Wei et al. (2025) | Distance Matrix | ✓ | ✗ | ✓ | ✗ | ✓ |
| **XDex (Ours)** | Joint Poses | ✓ | ✓ | ✓✓ | ✓ | ✓ |

Table 1: Dexterous Grasping Method Comparison.

that supports multi-objective grasping and generalizes to a large set of objects. DextrAH-G Lum et al. (2024b) and DextrAH-RGB Singh et al. (2024) incorporate geometric fabric into RL training, with the latter achieving successful sim-to-real transfer with RGB input. Our work aligns with the IL paradigm: we first generate a cross-embodiment grasp dataset via force-closure optimization and retargeting, and then train a unified grasping model on this dataset. We provide a detailed grasping method comparison in Tab. 1.

**Cross-embodiment Manipulation.** Cross-embodiment learning aims to train a unified policy that generalizes across different robot embodiments without retraining. One line of work adopts intermediate object-centric representations, such as contact points Shao et al. (2020); Attarian et al. (2023) and contact maps Li et al. (2023); Xu et al. (2024b); Morrison et al. (2018); Varley et al. (2015), followed by solving inverse kinematics to infer joint angles. While object-centric representations naturally support cross-embodiment generalization, they often suffer from limited accuracy due to the lack of robot hand understanding and are inefficient for joint value optimization. Another line of work incorporates robot hand representations into the model. NerveNet Wang et al. (2018) represents the robot structure as a graph to train a RL policy. UniGrasp Shao et al. (2020) learns a shared hand embedding space from point clouds using an autoencoder. ManiFM Xu et al. (2024b) and GeoMatch Attarian et al. (2023) directly input the hand's point cloud into the model. Body Transformer Sferrazza et al. (2024) and GET-Zero Patel & Song (2024) modify transformer attention masks based on joint graphs to encode hand morphology. The recent work D(R,O) Wei et al. (2025) encodes the interaction between hand and object using a dense point-to-point distance matrix, which is effective but computationally inefficient for learning and inference. In contrast, our method employs a lightweight joint-based representation and leverages a transformer model to handle heterogeneous joint structures.

**Hand Retargeting.** Hand retargeting is primarily used for teleoperation, where human hand poses are transformed into the joint positions of target robotic hands. Popular approaches include direct joint mapping Liu et al. (2017), supervised learning Li et al. (2019); Sivakumar et al. (2022), and energy-based optimization Handa et al. (2020); Qin et al. (2023b; 2022). CrossDex Yuan et al. (2025) utilizes a retargeting network for cross-embodiment RL policy training. Our method employs optimization-based retargeting to generate a paired dataset, and the trained model naturally supports environment-aware neural retargeting.

## 3 METHOD

### 3.1 OVERVIEW

The goal of cross-embodiment dexterous grasping is to generate physically plausible and diverse grasping poses that generalize across various objects and robotic hands. We leverage hand retargeting and grasp optimization techniques to construct a large-scale paired grasping dataset, in which each object is associated with multiple sets of grasping poses that are consistent across different hands. We then train a transformer-based generative grasping model and ensure consistency. The resulting model serves as a unified solution for both grasp synthesis and retargeting, capable of generating consistent, high-quality, and diverse grasps across different objects and hands. We further utilize this model to synthesize additional grasps, thereby expanding the dataset and further improving model performance. An overview of our proposed method is illustrated in Fig. 2.

### 3.2 PAIRED DATA GENERATION

To construct a paired grasping pose dataset across 1,000 hands, we need to address two challenges: (1) paired grasping pose generation; and (2) embodiment generation.

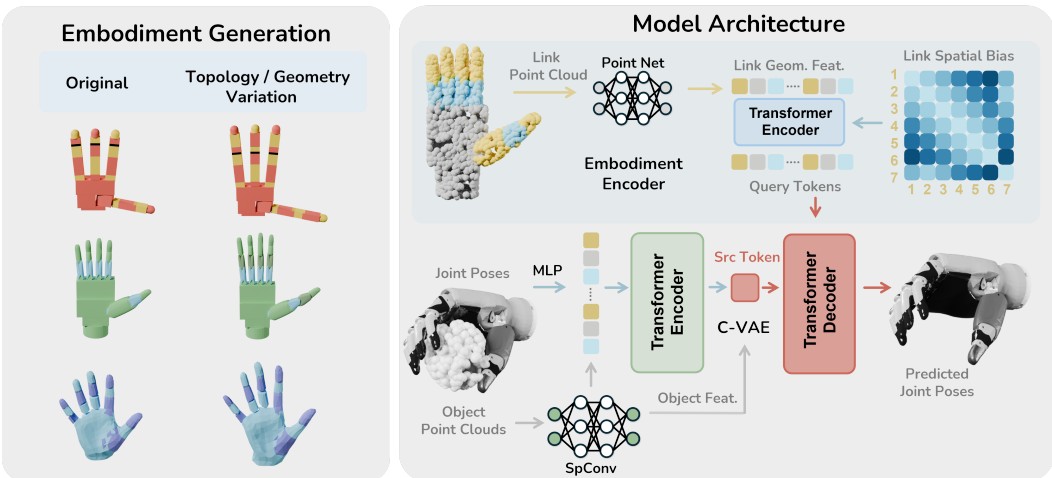

Figure 2: **Overview of XDex**. *Left:* We procedurally generate 1,000 diverse hand embodiments by varying topology and geometry, including the number of joints, link sizes, and shapes. *Right:* Given object and hand point clouds, XDex encodes hand geometry and topology together with object features. A transformer-based conditional variational autoencoder (CVAE) generates joint poses conditioned on a shared grasp feature and the input geometries, enabling consistent and diverse cross-embodiment grasp synthesis.

**Paired Grasping Pose Generation.** We first synthesize grasps for a *single source hand*, and then *retarget* the synthesized poses to a set of *target* hands. For each hand $h \in \mathcal{H}$, we denote a dexterous pose by $g^h = (T^h, R^h, \theta^h)$, where $T^h \in \mathbb{R}^3$ is the global translation, $R^h \in SO(3)$ the global rotation, and $\theta^h \in \mathbb{R}^{d_h}$ the joint angles of hand $h$ (e.g., $d_{\text{Ability}} = 6$, $d_{\text{Allegro}} = 16$). The object geometry is represented by its mesh $O$, while the hand geometry is approximated using a set of manually defined spheres (about 10 per link).

Let $h_{\text{s}}$ be the source hand. Following Wang et al. (2023); Liu et al. (2021), we obtain $g^{h_{\text{s}}}$ by minimizing the energy:

$$E_{\text{grasp}}(g^{h_{\text{s}}}) = E_{\text{fc}} + w_{\text{dis}}E_{\text{dis}} + w_{\text{sdf}}E_{\text{sdf}} + w_{\text{j}}E_{\text{j}} + w_{\text{s}}E_{\text{s}}, \tag{1}$$

where $E_{\text{fc}}$ enforces force closure, $E_{\text{dis}}$ reduces contact distance, $E_{\text{sdf}}$ penalizes penetration, $E_{\text{j}}$ keeps joints within limits, and $E_{\text{s}}$ avoids self-collision; $\{w_\star\}$ are scalar weights.

The penetration loss $E_{\text{sdf}}$ is based on the sphere approximation. Each sphere is parameterized by $(c, r)$, where $c \in \mathbb{R}^3$ is the center and $r \in \mathbb{R}^+$ is the radius. The center $c$ is transformed via forward kinematics. The SDF loss is defined as: $E_{\text{sdf}} = \sum_i r_i - \text{SDF}(c_i, O)$, where $\text{SDF}(c_i, O)$ denotes the signed distance from the sphere center $c_i$ to the object mesh $O$. To prevent collisions with the environment (e.g., the table), we take the maximum SDF across all scene meshes. The formal definitions of other terms follow prior works Wang et al. (2023); Liu et al. (2021). The optimized poses are passed into SAPIEN/ManiSkill simulators Xiang et al. (2020); Tao et al. (2024) to filter out physically implausible poses.

After optimizing the grasping pose for a *source* hand $h_{\text{s}}$, we retarget it to a *target* hand $h_{\text{t}} \neq h_{\text{s}}$ by solving an optimization problem that minimizes the distances between corresponding keypoints (e.g., fingertips and wrist) Qin et al. (2023b); Handa et al. (2020) and at the same time ensure the feasibility of the poses by minimize $E_{\text{sdf}}$ and $E_{\text{dis}}$ terms between the robot and the object:

$$\min_{g^{h_{\text{t}}}} \sum_\ell \left\| \text{FK}_\ell(g^{h_{\text{s}}}) - \text{FK}_\ell(g^{h_{\text{t}}}) \right\|^2 + w_{\text{sdf}} E_{\text{sdf}}^{h_{\text{t}}} + w_{\text{dis}} E_{\text{dis}}^{h_{\text{t}}}, \tag{2}$$

where $\text{FK}_\ell(\cdot)$ denotes the forward kinematics function that computes the world-frame position of the $\ell$-th keypoint. The terms $\text{FK}_\ell(g^{h_{\text{s}}})$ and $\text{FK}_\ell(g^{h_{\text{t}}})$ represent the corresponding keypoint positions of the source and target hands, respectively. All retargeted grasp poses are subsequently validated in simulation.

**Embodiment Generation.** To generate diverse robot hand embodiments, we consider two categories: human-like MANO hands Romero et al. (2017) and existing robot hands. We treat these

as source hands and employ a procedural pipeline to introduce both geometry and topology variations within each category. **Geometry Variations.** For MANO hands, we sample different shape parameters to generate diverse geometries. For robot hands, we either randomly scale the entire hand or randomly select a subset of links and independently scale them to introduce local variations. **Topology Variations.** For both MANO and robot hands, we apply the same set of operations: randomly duplicating or removing links, removing an entire finger, or merging two connected links into a single link. In total, we generate about 1,000 hands, including 200 MANO and 800 robot hands (see left side of Fig. 2). By procedurally generating both geometric and topological variations of MANO hands alongside robot hands, we unified human and robot embodiments into a common space, which expands the downstream applications, such as neural retargeting from human to robot and enables the capability of leveraging large-scale human data.

**Accelerated Paired Data Generation.** Since generating paired data solely through retargeting is computationally expensive, we randomly select only six hands per object for retargeting to initially train a reasonably performing model. This model is then used to generate paired data for more hands and objects, with the generated poses filtered by physics-based simulation to ensure physical plausibility. The new data are subsequently used to train a stronger model, which in turn produces more paired data, forming an iterative self-improving pipeline.

### 3.3 CROSS-EMBODIMENT GRASPING MODEL

We employ a transformer Vaswani et al. (2017)-based generative model for cross-embodiment grasping. The object geometry is represented by point cloud $P_O \in \mathbb{R}^{N_O \times 3}$. The robot hand geometry is represented by a set of link point clouds $\{P_\ell\}$, where each $P_\ell \in \mathbb{R}^{N_\ell \times 3}$ is in the wrist coordinate. The grasp pose is represented by the set of 6D Joint transformations $\{(R_\ell, T_\ell)\}$, where $R_\ell \in SO(3)$ and $T_\ell \in \mathbb{R}^3$ denote the rotation and translation of the parent joint of the $\ell$-th link, respectively. The model is trained using a conditional variational autoencoder (CVAE) Sohn et al. (2015), where the hand feature and object geometry serve as conditions. The architecture is detailed in the right side of Fig. 2.

**Geometry Feature Extraction.** We extract geometry features from both the robot hand and the object point clouds. For each robot link point cloud $P_\ell$, we apply PointNet Qi et al. (2017) to obtain a per-link feature vector $f_\ell^{\texttt{link}} \in \mathbb{R}^d$: $f_\ell^{\texttt{link}} = \texttt{PointNet}(P_\ell)$. For the object, we use SPConv Contributors (2022) to extract per-point features $f_p^{\texttt{obj}} \in \mathbb{R}^d$ for each point $p \in P_O$: $\{f_p^{\texttt{obj}}\}_{p \in P_O} = \texttt{SPConv}(P_O)$. These features serve as conditions for the CVAE-based transformer model: the link features $f_\ell^{\texttt{link}}$ and object features $f_p^{\texttt{obj}}$ capture the local geometry of each hand link and the object surface, respectively. Each grasping pose is associated with a single object point $p$ by projecting its heading direction onto the object surface, following DexGraspNet2.0 Zhang et al. (2024b). Hence, the local point features from SPConv are only used to condition a subset of local grasping poses (instead of all possible poses for the object), which helps the model learn better grasp consistency across different hands.

**Embodiment Encoder.** The geometric feature of each link is processed by a Transformer encoder. Meanwhile, the encoder learned attention bias in the self-attention mechanism to explicitly encode the topology information.Patel & Song (2025). Based on the URDF structure, we compute the shortest path distance (SPD) $\phi_{\text{SPD}}(i, j)$ between every pair of links. An embedding table $s$, indexed by this SPD value as $s_{\phi_{\text{SPD}}(i,j)}$, provides a learned scalar that is added to the attention score.

**Transformer-based CVAE.** We employ a transformer-based CVAE Sohn et al. (2015) to model the distribution of grasping poses. The encoder takes as input the grasp pose $g^h$ of a given robot hand $h$ and the local object geometry feature $f_p^{\texttt{obj}}$ at the corresponding point $p$, and encodes them into a latent distribution. The grasp pose $g^h$ is first converted into a set of link pose vectors $\{\pi_\ell = (R_\ell, T_\ell) \in \mathbb{R}^9\}^1$. Each link pose vector $\pi_\ell$ is concatenated with the corresponding object feature $f_p^{\texttt{obj}} \in \mathbb{R}^d$ to form a link token.

These link tokens are processed by a transformer encoder $\texttt{Enc}$ to produce per-token embeddings $\{e_\ell\}$, which are then aggregated via max pooling to obtain a global latent feature. A multi-layer perceptron (MLP) is used to map this feature to the mean $\mu$ and standard deviation $\sigma$ of the latent

---

[1]We adopt the 6D rotation representation Zhou et al. (2019), so each link pose is represented as a 9D vector.

distribution. A latent variable $z$ is then sampled using the reparameterization trick:

$$\{e_\ell\} = \texttt{Enc}(\{[\pi_\ell \,||\, f_p^{\texttt{obj}}]\}), \quad \mu, \sigma = \texttt{MLP}(\texttt{MaxPool}(\{e_\ell\})),$$

$$z = \mu + \sigma \odot \epsilon, \quad \epsilon \sim \mathcal{N}(0, I). \tag{3}$$

The sampled latent variable $z$ is concatenated with the object feature $f_p^{\texttt{obj}}$ to form a single source token for the transformer decoder. The decoder takes this token along with the link geometry features $\{f_\ell^{\texttt{link}}\}$ as query tokens to predict the joint poses:

$$\{\hat{\pi}_\ell\} = \texttt{Dec}(\{f_\ell^{\texttt{link}}\}, z \,||\, f_p^{\texttt{obj}}), \tag{4}$$

where each $\hat{\pi}_\ell \in \mathbb{R}^9$ represents a 6D rotation and a 3D translation for link $\ell$.

The model is trained using a combination of reconstruction loss and KL divergence:

$$\mathcal{L} = \lambda_{\texttt{R}} \sum_\ell \|\hat{\pi}_\ell - \pi_\ell\|^2 + \lambda_{\texttt{KL}} D_{\texttt{KL}}\left(\mathcal{N}(\mu, \sigma^2) \,\|\, \mathcal{N}(0, I)\right), \tag{5}$$

where $\hat{\pi}_\ell$ is the predicted pose for link $\ell$, and $\pi_\ell$ is the ground truth.

Finally, we solve an inverse kinematics (IK) problem to recover the hand pose $\hat{g}^h$ that matches the predicted joint poses:

$$\hat{g}^h = \arg\min_{g^h} \sum_\ell \left\|\texttt{FK}_\ell(g^h) - \hat{\pi}_\ell\right\|^2, \tag{6}$$

where $\texttt{FK}_\ell(g^h)$ computes the pose of link $\ell$ from the hand pose $g^h$.

**Encouraging Consistency.** While the paired dataset (see Sec. 3.2) implicitly encourages consistent grasping poses across different hands given the same object feature $f_p^{\texttt{obj}}$ and latent variable $z$, we further introduce a *retargeting loss* to enforce consistency.

We assign the same sampled latent variable $z$ and object feature $f_p^{\texttt{obj}}$ to two different hands, $h_1$ and $h_2$, with corresponding link geometry features $\{f_\ell^{\texttt{link}}(h_1)\}$ and $\{f_\ell^{\texttt{link}}(h_2)\}$. The transformer decoder predicts joint poses $\{\hat{\pi}_\ell^{h_1}\}$ and

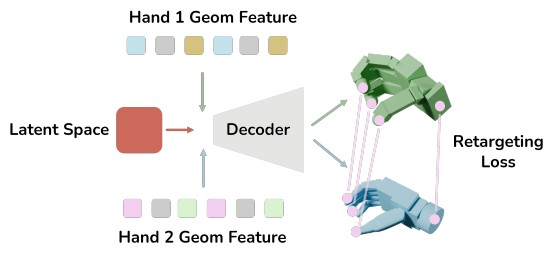

Figure 3: **Consistency Loss Terms**. Shared latent grasp feature is decoded into poses for different hands, with a retargeting loss to enforce consistency.

$\{\hat{\pi}_\ell^{h_2}\}$ for the two hands. The retargeting loss penalizes the difference between the corresponding keypoint positions: $\mathcal{L}_{\texttt{retarget}} = \sum_\ell \left\|\hat{\pi}_\ell^{h_1} - \hat{\pi}_\ell^{h_2}\right\|^2$.

The final loss is a combination of loss terms described above. The proposed consistency loss term is illustrated in Fig. 3.

**Neural Retargeting.** Our grasp synthesis model naturally supports object-conditioned neural retargeting. Given a set of joint poses $\{\pi_\ell^{h_1}\}$ from a source hand $h_1$ and the shared object feature $f_p^{\texttt{obj}}$, we first encode the input using the transformer encoder to obtain the latent mean $\mu$ (without sampling). We then decode this latent mean using the link geometry features $\{f_\ell^{\texttt{link}}(h_2)\}$ of a target hand $h_2$ to predict the corresponding joint poses:

$$\{\hat{\pi}_\ell^{h_2}\} = \texttt{Dec}(\{f_\ell^{\texttt{link}}(h_2)\}, \mu \,||\, f_p^{\texttt{obj}}), \tag{7}$$

where $\mu = \texttt{MLP}(\texttt{MaxPool}(\texttt{Enc}(\{[\pi_\ell^{h_1} \,||\, f_p^{\texttt{obj}}]\})))$.

## 4 EXPERIMENTS

We evaluate the quality, consistency, and diversity of the grasp synthesis results in Sec. 4.2. We then study the effect of scaling the amount of embodiments in Sec. 4.3. Real-world deployment results are reported in Sec. 4.4.

| Method | Quality | | | | Consistency | | | | Diversity | | | |
|---|---|---|---|---|---|---|---|---|---|---|---|---|
| | SR ↑ | | Pen ↓ | | Pos Diff ↓ | | Vec Diff ↓ | | H mean ↑ | | H std ↓ | |
| | Seen | Unseen | Seen | Unseen | Seen | Unseen | Seen | Unseen | Seen | Unseen | Seen | Unseen |
| Retargeting (wo filter) | 0.35 | 0.46 | 0.17 | 0.18 | 0.42 | **0.49** | 0.20 | 0.21 | 3.91 | 3.64 | 0.02 | 0.02 |
| D(R,O)* Wei et al. (2025) | 0.54 | - | 0.21 | - | 0.42 | - | 0.15 | - | 3.68 | - | 0.03 | - |
| Ours | **0.77** | **0.72** | **0.15** | 0.19 | **0.42** | 0.52 | **0.18** | 0.22 | 3.89 | 3.67 | 0.11 | 0.07 |
| Ours w/o paired data | 0.50 | 0.23 | 0.20 | 0.17 | 0.46 | 0.58 | 0.18 | 0.22 | 3.72 | 3.84 | 0.03 | 0.05 |
| Ours w/o embodiment encoder | 0.72 | 0.68 | 0.17 | 0.20 | 0.44 | 0.50 | 0.22 | 0.23 | 3.61 | 3.67 | 0.03 | 0.07 |
| Ours with consistency loss | 0.77 | 0.64 | 0.18 | 0.17 | 0.42 | **0.50** | 0.18 | **0.21** | 3.55 | 3.63 | 0.04 | 0.07 |

Table 2: **Grasping synthesis results.** We report grasp quality, consistency, and diversity metrics on both seen and unseen robot hands. Our method outperforms baselines and ablated variants across most metrics, especially on unseen hands.

## 4.1 EXPERIMENTAL SETTINGS

**Data.** We collect object assets from multiple sources: DexGraspNet Wang et al. (2023), EGAD Morrison et al. (2020), RoboCasa Nasiriany et al. (2024), KIT Kasper et al. (2012), ContactDB Brahmbhatt et al. (2019), and YCB Calli et al. (2015). The test split includes all objects from KIT, YCB, and ContactDB, as well as the test sets from EGAD and DexGraspNet. All other objects are included in the training split. All objects are convex decomposed and normalized. Since we focus on a tabletop grasping setting, objects must be stably placed on the table plane during simulation. We randomly sample the translation, orientation, and scale of each object and run simulations to determine stable poses. In total, we generate 70K scenes for training and 500 scenes for testing.

For hand assets, we generate a total of 1,000 hand embodiments. Specifically, we create 200 variations of the human-like MANO hand by sampling diverse shape parameters. The remaining 800 variations are procedurally generated based on five robot hands: Allegro, Inspire, Ability, Schunk, and Shadow. To evaluate generalization, we keep the original Inspire, Ability, Schunk, and Shadow hands as unseen hands for testing. For seen hands, we randomly select 20 variations for evaluation.

**Metrics.** We categorize our evaluation metrics into three groups: Quality (Success Rate, Penetration), Consistency (Position difference, Vector difference), and Diversity (H mean, H std). A grasp is considered successful if the hand starts from a pre-grasp pose (no contact with the object), lifts the object above a $0.2\,\text{m}$ threshold, and maintains contact with the object using at least two fingertips. While the quality and diversity metrics are commonly used in prior works Wang et al. (2023); Lu et al. (2024), we introduce consistency metrics to evaluate grasp pose similarity across different hands. These two metrics are adapted from energy functions used in previous retargeting works Qin et al. (2023b): *position difference* is the average L2 distance between the fingertips and wrist in the world coordinate, while *vector difference* is the average L2 distance between the fingertips in the wrist coordinate. The former emphasizes global differences, whereas the latter focuses more on the grasping pose. For multi-hand evaluations, the reported value is the average difference across all possible grasp pairs between two different hands.

**Baselines.** We mainly compare our method against two baselines: (1) *Retargeting*, where a grasping policy is trained on a single hand and retargeted to other hands during evaluation; and (2) *D(R,O)*, a state-of-the-art method that conditions on the robot hand point cloud and predicts the distance between the hand and the object for cross-embodiment grasping. Since D(R,O) is designed for grasping floating objects, we re-train their model on our tabletop grasping dataset and denote this adapted version as D(R,O)*. We fix their pre-trained robot encoder during training, as it is crucial to final performance Wei et al. (2025). We also observe that the batch size of D(R,O)* is limited to 4 due to the large memory requirement of the dense distance matrix.

## 4.2 GRASPING SYNTHESIS RESULTS

We evaluate grasp synthesis performance across three axes: *Quality*, *Consistency*, and *Diversity*. Table 2 reports detailed metrics on both seen and unseen hands. Our method outperforms both prior baselines and ablated variants across most metrics.

**Quality.** Our method outperforms both the retargeting baseline and D(R,O)* in terms of success rate and penetration. On unseen hands, we achieve a success rate of 0.77, compared to 0.57 for

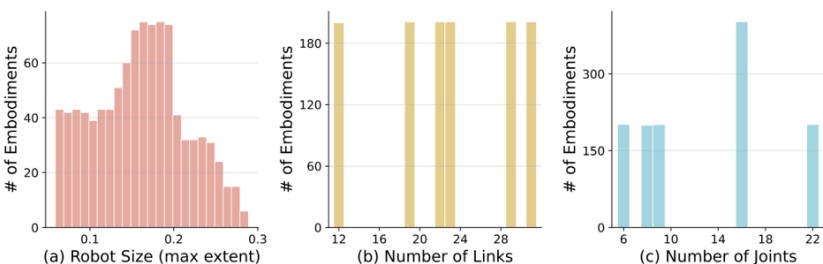

Figure 5: Distribution and scaling of hand embodiments used in XDex. (a)-(c): Our 1,000 generated robot hands cover a wide range of sizes, link counts, and joint counts, capturing diverse morphologies

retargeting. The performance of D(R,O)* is worse than the retargeting baseline, and we even fail to generalize D(R,O)* to unseen hands. We conjecture that this is because our dataset contains much higher embodiment and grasping diversity than the original setting used to pre-train D(R,O). As we fix the pre-trained hand encoder during re-training, it may not generalize well to our dataset.

**Diversity.** Our method achieves comparable diversity compared to baselines, with an entropy mean of 3.67 and entropy std of 0.07 on unseen hands. This suggests that our model not only produces consistent grasps but also maintains sufficient diversity.

**Disentanglement Visualization.** To better understand the behavior of our model, we visualize the disentangled grasping poses for a single object in Fig. 6. Each row shows grasping poses generated from the same object point $p$ and the same grasp feature $z$, but for different hand embodiments. Our method produces consistent and physically plausible grasps across hands, demonstrating successful disentanglement of object, hand, and grasp representations. Moreover, different rows show various grasping poses for the same object, indicating the pose diversity of our method.

**Ablation Study.** We evaluate the effectiveness of using *paired data* and *embodiment encoder* and report the results in Tab. 2. Instead of pairing grasping poses across hands, we independently optimize for each hand and train the model with this mixed data. This leads to a significant drop in performance, especially on unseen hands (e.g., the success rate drops from

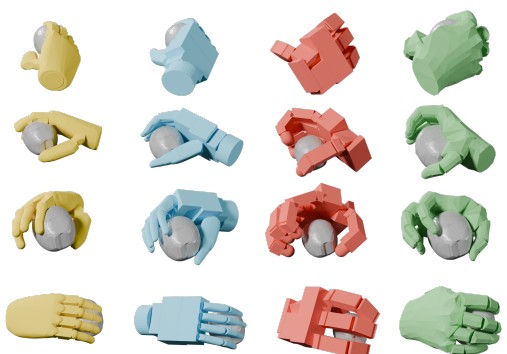

Figure 4: Disentangled grasping poses for one object across multiple object points and hand embodiments. Each **row** shows consistent grasps across different hands for the same object point and grasp feature, while different rows illustrate diverse grasping strategies for different object points.

0.72 to 0.23). This result highlights the importance of cross-embodiment grasping pose pairing, as it enables the model to better focus on subtle morphological differences for the same object point and grasp features. Using the embodiment encoder improves the success rate by 5% on seen hands and 4% on unseen hands compared to not using it, demonstrating that encoding topological information helps the model better understand the embodiment structure and thus enhances its performance.

**Inference Efficiency.** We evaluate the inference speed of our method compared to baselines. XDex requires approximately 55 ms per grasp prediction, including both the network forward pass and post-processing. In contrast, DR-O retargeting methods typically take around 500 ms per grasp, demonstrating that our method is better suited for real-time applications.

### 4.3 EMBODIMENT SCALING LAWS

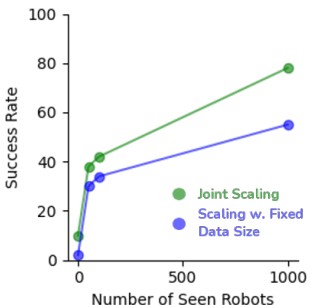

Figure 7: Zero-shot deployment of our method on an xArm with an Ability Hand. Our model directly transfers to the real world from simulation, successfully executing open-loop grasps across a variety of everyday objects using point cloud input.

We generate 1,000 diverse robot hand embodiments covering different sizes, numbers of links, and numbers of joints, and report the statistics in Fig. 5. Leveraging these embodiments, We investigate the scaling law in two settings: (i) increasing the number of embodiments while keeping the total data size fixed—so that the data per embodiment decreases as the number of embodiments grows, and (ii) jointly scaling both the data size and the number of embodiments.

**Scaling Embodiment with Fixed Data Size.** While keeping the number of object assets and grasping poses fixed, increasing the number of seen embodiments improves the model's success rate on unseen hands, but the performance eventually saturates at around 50%.

**Joint Scaling of Data Size and Embodiment.** We further investigate the scaling law by jointly scaling both the data size and the number of embodiments. We observe a much steeper improvement in grasping success rate as more embodiments and more data are incorporated during training, with performance on unseen hands eventually reaching 77%, highlighting the importance of embodiment diversity and data diversity for cross-embodiment generalization.

Figure 6: Scaling law analysis shows that increasing the number of seen embodiments and the data size during training improves success rate, indicating the importance of embodiment diversity and data size during cross-embodiment grasping learning

### 4.4 REAL-WORLD DEMONSTRATION

We demonstrate the effectiveness of the proposed method in the real world through direct sim-to-real deployment. We use an xArm with an Ability Hand platform and mount a camera in a third-person view. The camera pose is calibrated using standard hand-eye calibration. The partial point cloud observation from the camera is taken as input to the model.

To execute a grasp, we sample 128 candidate grasp poses and solve inverse kinematics (IK) to select a feasible one. The final grasp pose is executed in an *open-loop* manner. Both the reaching and lifting motions are achieved via motion planning based on the selected pose. Grasping poses are visualized in Fig. 7.

## 5 CONCLUSION

We present **XDex**, a cross-embodiment grasp synthesis framework that disentangles object geometry, hand embodiment, and grasping pose to achieve generalizable and co nsistent dexterous grasping across 1,000 robot hands. By constructing a large-scale paired grasping dataset and leveraging a transformer-based conditional variational autoencoder, XDex is able to generate high-quality, consistent, and diverse grasps for both seen and unseen embodiments. We introduced explicit retargeting and embedding losses to further enforce grasping consistency across embodiments, and demonstrated that our approach significantly outperforms strong baselines across multiple metrics. Our experimental results confirm the importance of embodiment diversity and disentangled representation in enabling cross-embodiment generalization. Furthermore, XDex supports real-world sim-to-real deployment. We believe XDex provides a promising step towards foundation models for dexterous manipulation that generalize across robotic hands. We are committed to releasing the code upon acceptance.

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

## A   APPENDIX

**Use of Large Language Model.** We use LLM to aid or polish writing.