# OpenReview forum: "XDex: Learning Cross-Embodiment Dexterous Grasping with 1000 Hands"
_ICLR.cc/2026/Conference — ICLR 2026 Conference Withdrawn Submission_

### Official Review · Reviewer_xWAV · 2025-10-26

**Soundness:** 3
**Presentation:** 3
**Contribution:** 4
**Rating:** 6
**Confidence:** 4

**Summary:**

This paper proposes XDex, a large-scale cross-embodiment grasping dataset. The dataset is constructed by first generating grasps for a source human hand and then adapting them to multiple robotic hands and their variations. By training on this dataset, a model can learn to generate grasps given object and hand features, enabling generalization to diverse hand structures.

**Strengths:**

1.  The paper proposes learning cross-embodiment grasp generation for various robotic hands from scalable synthetic data. The core idea of generating hand variations and retargeting from a human hand is novel and interesting.
2.  The analysis of grasp consistency across different hands is an interesting point. I would encourage a more detailed analysis of this, such as probing the similarities between the hand features of corresponding links.
3.  The evaluations are detailed, revealing promising quality in the generated grasping poses, and the real-world deployment results are appreciated. The results provide insights into achieving scaling laws for cross-embodiment robotic grasping.
4.  The paper is well-written and easy to read, and the visualizations are informative.

**Weaknesses:**

1.  More details on the generation of embodiments (e.g., shape parameter distributions, mechanisms to merge or add links) should be provided. According to Fig. 5, it seems that the link and joint numbers are sparsely distributed, which may limit data diversity.
2.  The grasp execution evaluation could be improved. Since the grasp generation method ensures Force Closure, it is more common to evaluate the grasp by applying gravity from multiple directions, rather than just a single one.

**Questions:**

1.  How could this pipeline generalize to non-humanoid hands, such as the D'Claw or parallel grippers, which have significant morphological variations and lack a clear unified source like the human hand?
2.  While Fig. 5 shows the distribution of multiple metrics, I am curious about the distribution of the general hand morphologies. A t-SNE visualization of the hand geometry features or a similar technique would be insightful.

---

### Official Review · Reviewer_9bqW · 2025-10-28

**Soundness:** 2
**Presentation:** 2
**Contribution:** 3
**Rating:** 2
**Confidence:** 4

**Summary:**

The authors propose XDex, which addresses the challenge of generalizing dexterous grasping across diverse robotic hands. The authors hypothesize that scaling the diversity of hand embodiments is key to generalization. XDex is trained on a new, large-scale dataset of 1,000 procedurally generated hand embodiments with paired grasping poses. It employs a transformer-based CVAE architecture with an embodiment encoder that captures both hand geometry and topology, along with an explicit retargeting loss. Experiments show that XDex significantly outperforms prior methods on both seen and unseen hands in grasp quality and consistency in a zero-shot manner.

**Strengths:**

- The authors' central premise of scaling embodiment diversity to improve generalization is compelling. The creation of a dataset with 1,000 diverse hand embodiments is a major contribution in itself. The strong performance on unseen hands is proof of generalization.
- The concept of a paired grasping dataset combined with an explicit retargeting loss is an effective way to enforce grasp consistency. The embodiment transformer jointly encoding per-link geometry (via PointNet) and kinematic topology (via attention bias from shortest path distance) is a novel way to represent diverse robot morphologies.
- The experimental validation is extensive and uses a well-defined set of metrics. The successful zero-shot sim-to-real deployment further demonstrates robustness.

**Weaknesses:**

While the paper presents an interesting and novel idea, the paper is far from reaching publication quality in its current form:
- The most significant weakness of this paper is the complete absence of an appendix. The 9-page limit is insufficient to cover the necessary details. This major flaw makes it impossible to fully assess the work and prevents other researchers from building upon it, especially in a venue like ICLR, which embraces reproducibility. Specifically, the final pose generation step relies on solving an inverse kinematics problem (Equation (6)), but all practical details are omitted. The paper does not specify the solver used, the initialization strategy, how joint limits are handled, or the procedure for dealing with non-convergence. Moreover, the procedural generation of the 1,000-hand dataset is a core contribution, yet its description lacks detailed information. The paper does not state the constraints or heuristics used during "Topology Variations" to ensure the generated kinematic chains are physically plausible. There is no justification for the choice of the five base robot hands, nor are there qualitative examples of the more unusual or challenging morphologies generated, which would help in understanding the dataset's true diversity and potential biases. Details about the "Retargeting" baseline are also underspecified. The paper does not state which specific single-hand grasping policy was trained or which retargeting algorithm was used for evaluation.
- The submission includes neither an Ethics Statement nor a Reproducibility Statement. The authors also fail to include a discussion of hyperparameter choices, computation usages, detailed dataset statistics, and efficiency reports, which raises further concerns about reproducibility.
- The paper does not discuss several highly relevant and recent works on dexterous grasp transfer/retargeting, such as RobotFingerPrint (https://irvlutd.github.io/RobotFingerPrint/), FunGrasp (https://hly-123.github.io/FunGrasp/), Dexonomy (https://pku-epic.github.io/Dexonomy/), and Grasp2Grasp (https://grasp2grasp.github.io/). While some may be concurrent under the ICLR policy, acknowledging and briefly differentiating from them would significantly strengthen the paper's contribution claims.
- Minor: The manuscript completely misuses `\citep` vs. `\citet`.

**Questions:**

See weaknesses.

---

### Official Review · Reviewer_8yDh · 2025-11-01

**Soundness:** 3
**Presentation:** 3
**Contribution:** 3
**Rating:** 6
**Confidence:** 3

**Summary:**

The authors introduce a cross embodiment grasp dataset, and grasp synthesis pipeline explicitly designed to leverage embodiment information. Because embodiment information is explicitly encoded, the proposed model can effectively transfer synthesized grasps between hand morphologies. The authors make the following claims: (1) scaling the diversity and number of embodiments improves generalization to unseen hands, (2) incorporating embodiment information improves quality, consistency, and diversity. The former is not shown, but the latter is.

**Strengths:**

**Originality.** The work appears original.

**Quality.** The work appears to be of high quality. The work is well motivated, the design decisions are consistent with the motivation and hypotheses, and (for the most part) the experiments validate the claims.

**Clarity.** The paper is well written and well organized.

**Significance.** The experiments demonstrate that using a large dataset improves performance. While this result is not particularly surprising, the introduction of such a large dataset carries significance. The experiments also demonstrate that the proposed architecture (designed to explicitly encode embodiment) leads to improved grasp synthesis. This also carries significance especially since the authors have shown this architecture can be used to enlarge the dataset.

**Weaknesses:**

My main critique of the paper is that it doesn’t seem that the claim: increasing the number of embodiments improves generalization to unseen hands, has been shown. In Section 4.3, the authors show: scaling embodiment with a fixed data size, and joint scaling of data size with embodiment. Adding the experiment: scaling data size with fixed number of embodiments would resolve my concern.

**Questions:**

**Questions**

- Line 251: the local point features from SPConv are only used to condition a subset of local grasping poses
- In eq 4 the decoder is conditioned on f^{obj}_p which is encoded in z. Is this necessary?
- In eq 5 and 6 is the distance between poses the Euclidean distance? This metric doesn’t respect the geometry of the rotation group. Do you have a sense of the implications of this?
- “We also observe that the batch size of D(R,O)* is limited to 4 due to the large memory requirement of the dense distance matrix” How does your proposed method compare?
- What happens when the data size grows and the embodiments don't?

**Possible typos**

- Line 238: 6D Joint → 6D joint
- Line 256: information. → information
- Line 255: Meanwhile, the encoder learned attention bias in the self-attention mechanism to explicitly encode → The encoder learned attention bias in the self-attention mechanism is used to explicitly encode
- Line 396: Our method achieves comparable diversity compared to baselines → Our method achieves diversity comparable to baselines
- Line 476:  co nsistent → consistent
- Line 332: Ours with consistency → Ours w/o consistency

---

### Official Review · Reviewer_2jri · 2025-11-02

**Soundness:** 3
**Presentation:** 3
**Contribution:** 2
**Rating:** 4
**Confidence:** 4

**Summary:**

This paper proposes XDex, a framework for generating dexterous grasps that can generalize across a large set of 1,000 different robot hands. The method trains a transformer-based conditional VAE on a large-scale paired dataset, where grasps are retargeted across embodiments to enforce consistency, enabling generalization to unseen hands.

**Strengths:**

1. The paper introduces a large-scale paired grasping dataset covering 1,000 hand embodiments.
2. The model architecture encodes hand topology using an attention bias, which is an interesting way to handle varied kinematics.
3. The experimental analysis that separates the effects of data size and embodiment count is a good analysis.

**Weaknesses:**

1. The experimental setup for "unseen hands" does not convincingly support the claim of generalization, as the test hands are the base models used to procedurally generate the entire training set.
2. The paper lacks a qualitative analysis of failure modes or a discussion of limitations, which would be crucial for understanding the boundaries of the method's capabilities.
3. The effect of the consistency loss is vague. The Table. 2 shows that "Ours with consistency loss" is comparable to "Ours". Based on this result, I believe authors should discuss more on this loss and why they add it.

**Questions:**

1. Regarding the experiment in Sec 4.1, the unseen test hands (Inspire, Ability, etc.) are the original models from which the 800 training hands are procedurally generated. Could you clarify how this setup truly tests generalization to out-of-distribution hand morphologies, rather than just interpolation within a known distribution?
2. What are the primary failure modes of the model?
3. The consistency loss ablation in Table 2 shows limited change in metrics. Can you elaborate more on how this loss influence the results?
4. The data generation process is described as "computationally expensive." Could you provide a quantitative figure for the time required to generate the full paired dataset? This information is important for assessing the practicality and reproducibility of the data pipeline.
5. Could you provide more detail on the dataset, such as the number of unique object models in the test split?

---

### Note · Authors · 2025-11-14

I have read and agree with the venue's withdrawal policy on behalf of myself and my co-authors.